# Implementation of an SS-Compensated LC-Thermistor Topology for Passive Wireless Temperature Sensing

**DOI:** 10.3390/s25206316

**Published:** 2025-10-13

**Authors:** Seyit Ahmet Sis, Yeliz Dikerler Kozar

**Affiliations:** Electrical and Electronics Engineering, Balikesir University, Balikesir 10145, Turkey; yelizdikerler@gmail.com

**Keywords:** passive wireless sensor, LC-thermistor, SS-compensated topology, bifurcation

## Abstract

**Highlights:**

**What are the main findings?**

**What is the implication of the main finding?**

**Abstract:**

This paper presents a passive wireless temperature sensor based on an SS-compensated LC-thermistor topology. The system consists of two magnetically coupled LC tanks—each composed of a coil and a series capacitor—forming a series–series (SS) compensation network. The secondary side includes a negative temperature coefficient (NTC) thermistor connected in series with its coil and capacitor, acting as a temperature-dependent load. Magnetically coupled resonant systems exhibit different coupling regimes: weak, critical, and strong. When operating in the strongly coupled regime, the original resonance splits into two distinct frequencies—a phenomenon known as bifurcation. At these split resonance frequencies, the load impedance on the secondary side is reflected as pure resistance at the primary side. In the SS topology, this reflected resistance is equal to the thermistor resistance, enabling precise wireless sensing. The advantage of the SS-compensated configuration lies in its ability to map changes in the thermistor’s resistance directly to the input impedance seen by the reader circuit. As a result, the sensor can wirelessly monitor temperature variations by simply tracking the input impedance at split resonance points. We experimentally validate this property on a benchtop prototype using a one-port VNA measurement, demonstrating that the input resistance at both split frequencies closely matches the expected thermistor resistance, with the observed agreement influenced by the parasitic effects of RF components within the tested temperature range. We also demonstrate that using the average readout provides first-order immunity to small capacitor drift, yielding stable readings.

## 1. Introduction

Wireless passive sensors are widely employed in critical applications with battery free interrogation and easy embedding in constrained environments. In biomedical contexts, passive tags support in situ monitoring where size and safety constraints are paramount [1,2,3,4,5,6]. For operation in harsh environments—e.g., high-temperature industrial settings and high-pressure systems, passive approaches offer robustness where wired sensors or on-board power are impractical [7,8,9]. In structural-health applications, passive sensors have been deployed for in situ bearing health monitoring, for high-temperature corrosion monitoring of metals, and for bearing temperature sensing applications—highlighting their suitability for long-term, distributed measurements over large areas [10,11,12]. Related developments in industrial process control further demonstrate the practicality of passive temperature and pressure monitoring in everyday systems [13,14,15,16,17].

Among various designs, LC (inductor–capacitor) topology is the most prevalent due to its cost-effectiveness and ease of implementation. Typically, the sensing side—referred to as the secondary side—comprises a capacitor usually sensitive to the target physical parameter, connected in series with an inductor. A corresponding primary coil, positioned in proximity to the sensor coil, constitutes the reader side of the system. This arrangement enables contactless sensing through magnetic coupling between the primary and secondary coils.

Although capacitors are most commonly used as the sensing element in wireless passive sensor architectures due to their straightforward integration and high sensitivity to environmental changes, alternative configurations have also been explored in the literature. In certain applications, the sensing functionality is instead provided by the inductor (coil) [18], the load resistor [19,20,21], or even the mutual inductance [22,23] between the coupled coils, which reflects the degree of magnetic coupling and varies with environmental or spatial parameters. These alternative sensing elements offer unique advantages depending on the nature of the physical parameter being monitored and the design constraints of the system. Regardless of which element is employed as the sensing component, any change in its value—caused by the variation in the measured physical quantity—results in a shift in the overall input impedance and a corresponding change in the resonance frequency of the coupled resonant circuit.

Traditionally, the primary side of a wireless LC sensor system, where the readout operation is conducted, consists solely of a single primary coil, which also serves as the defined input port. However, recent developments in wireless LC sensors have increasingly focused on enhancing sensitivity, expanding readout range, and enabling multi-parameter detection. As stated in [24], the inclusion of resonant circuitry on the primary side has led to architectures resembling magnetically coupled resonator systems, commonly used in modern wireless power transfer (WPT) applications. This approach improves both sensing performance and system integration by enabling the use of analytical tools and design strategies from the WPT domain. Building upon this concept, Zhou et al. [25] demonstrated that multi-parameter measurement can be realized using PT-symmetric dual-resonator systems, where variations in capacitance, resistance, or mutual inductance cause detectable shifts in reflection response near exceptional points (EPs). Chen et al. [26] introduced a generalized parity-time symmetric condition in RF telemetry systems with enhanced resolution and sensitivity. Dong et al. [27] advanced this concept by implementing an EP-locked wireless readout system for implantable LC microsensors.

Hajizadegan et al. [28] presented a PT-symmetric displacement sensing architecture where EP-based eigenfrequency bifurcation was exploited for high-sensitivity remote sensing. Their system enabled submillimeter displacement tracking without physical contact, using only low-cost printed coils. Zhou et al. [29] also investigated PT-symmetry breaking in LC sensors, showing that operating under a PT-asymmetric regime could effectively increase the sensing distance between the readout and sensor coils. Their approach mitigates the limitations of weak magnetic coupling, particularly relevant in sealed, miniaturized, or implantable environments. Takamatsu et al. [30] developed wearable and implantable LC bioresonators based on amplitude-modulated PT-symmetric telemetry. Their system achieved more than 2000-fold improvement in signal sensitivity and a 78% reduction in detection error for biochemical sensing of glucose and lactate concentrations, demonstrating robustness and suitability for low-power, low-volume wearable applications.

While prior studies on PT-symmetric or both-side resonant LC sensor systems have primarily emphasized eigen frequency bifurcation, enhanced sensitivity, and multi-parameter capabilities, an essential feature intrinsic to such coupled resonator configurations—especially under series-series (SS) compensation—is often overlooked. In SS-compensated topologies, where both the reader and sensor coils are connected in series with capacitors, a key advantage emerges at the bifurcation (or split resonance) condition: the load resistance on the sensor side is precisely reflected to the input port as an equivalent input impedance [31,32,33,34,35]. This property allows for the direct and wireless readout of resistive sensing elements—such as thermistors—without the need for active electronics, making the topology exceptionally suitable for compact, battery-free, and contactless resistive sensing applications.

In this study, we exploit this characteristic by implementing a fully passive wireless temperature sensing system based on an SS-compensated LC-thermistor configuration. The sensing platform consists of two magnetically coupled and identical LC resonators, each comprising a coil and a series capacitor to form the SS topology. On the sensing side, a negative temperature coefficient (NTC) thermistor is connected in series with its LC tank, serving as a temperature-dependent resistive load. The system enables the wireless interrogation of temperature through the reflected impedance observed at the input port, leveraging the impedance mirroring behavior unique to the SS-compensated structure.

The novelty of this work lies in the first practical implementation of the SS-compensated topology for wireless resistive sensing exploiting the direct reflection of sensing resistor to the reader side. While previous studies have mainly employed SS or PT-symmetric configurations for wireless power transfer or theoretical demonstrations, our study demonstrates that the impedance mirroring effect inherent to SS compensation can be directly exploited for battery-free, and contactless temperature measurement.

Compared with mainstream approaches, the SS-compensated architecture reduces reader complexity and cost while remaining fully passive. Frequency-shift LC sensors infer temperature from the resonance shift in a single tank and therefore require a frequency sweep circuit such as PLL, which complicates and raises the cost of the reader. RFID-based solutions embed an IC and protocol stack, adding overhead and relying on RF-field power. By contrast, our implementation operates at the split zero-phase-angle frequencies $f_{L}$ and $f_{H}$, where the thermistor resistance is mirrored as a pure input resistance; thus, a single-/two-tone narrowband reader suffices. Relative to active, transceiver-based sensors, the proposed sensor is battery-free. Taken together direct resistance mapping at $f_{L},\;f_{H}$, an IC-free tag, and a sweep-free reader constitute the core innovation, providing a sensitive, low-cost, and structurally simple alternative for passive wireless temperature sensing.

## 2. Mathematical Modeling of the SS-Compensated LC-Thermistor Sensor Topology

Figure 1 illustrates the circuit model of the SS-compensated LC-Thermistor topology, which consists of two magnetically coupled resonant tanks. On the primary (reader) side, an inductor *L_p_* is connected in series with a capacitor *C_p_*, forming a series LC circuit. On the secondary (sensor) side, another series LC circuit is formed by an inductor *L_s_* and a capacitor *C_s_*, which is loaded by a resistive sensing element *R_L_*. The mutual inductance *M* between the coils *L_p_* and *L_s_* enables interaction between the reader and sensor circuits. Variations in *R_L_*, due to temperature changes, influence the input impedance *Z_in_* observed at the primary side, enabling passive and contactless temperature monitoring.

The input impedance ($Z_{in}$) of the SS-compensated LC sensor topology can be expressed as:(1)$$Z_{in\;}=\frac{1}{j\omega C_{p}}+j\omega L_{p}+Z_{r}$$Here, $Z_{r}$ denotes the impedance reflected from the secondary side to the primary, accounting for the sensor’s loading effect [35]. It is derived as:(2)$$Z_{r}=\frac{\omega^{4}C_{s}^{2}M^{2}R_{L}}{{\left(\omega^{2}C_{s}L_{s}-1\right)}^{2}+\omega^{2}C_{s}^{2}R_{L}^{2}}-j\frac{\omega^{3}C_{s}M^{2}\left(\omega^{2}C_{s}L_{s}-1\right)}{{\left(\omega^{2}C_{s}L_{s}-1\right)}^{2}+\omega^{2}C_{s}^{2}R_{L}^{2}}$$By substituting Equation (2) into Equation (1), a full expression for the input impedance is obtained:(3)$$Z_{\mathrm{in}}=\frac{C_{s}^{2}M^{2}R_{L}\omega^{4}}{C_{s}^{2}R_{L}^{2}\omega^{2}+{\left(C_{s}L_{s}\omega^{2}-1\right)}^{2}}+j\left(-\frac{1}{C_{p}\omega }+L_{p}\omega -\frac{\omega^{3}C_{s}M^{2}\left(\omega^{2}C_{s}L_{s}-1\right)}{{\left(\omega^{2}C_{s}L_{s}-1\right)}^{2}+\omega^{2}C_{s}^{2}R_{L}^{2}}\right)$$To ensure optimal signal transfer between primary and secondary side, the secondary circuit is tuned to its resonance frequency:(4)$$\omega_{0}=\frac{1}{\sqrt{C_{s}L_{s}}}$$Furthermore, in order to achieve a purely resistive input at resonance, the primary side reactance is compensated by appropriately selecting the capacitor $C_{p}$ as:(5)$$C_{p}=\frac{1}{\omega_{0}^{2}L_{p}}$$When Equations (4) and (5) are enforced, the input impedance expression in Equation (3) becomes more manageable, and the resonance frequencies at which the input reactance vanishes (i.e., zero-phase angle or ZPA points) can be found by setting the imaginary part of $Z_{in}$ to zero [31]:(6)$$Im\left(Z_{in}\right)=\frac{-1+\omega^{2}/\omega_{0}^{2}}{C_{p}\omega }-\frac{C_{s}M^{2}\omega^{3}\left(-1+\frac{\omega^{2}}{\omega_{0}^{2}}\right)}{C_{s}^{2}R_{L}^{2}\omega^{2}+{\left(-1+\frac{\omega^{2}}{\omega_{0}^{2}}\right)}^{2}}=0.$$Equation (6) can further be factorized as(7)$$Im\left(Z_{in}\right)=\left(-1+\frac{\omega^{2}}{\omega_{0}^{2}}\right)\left[\frac{1}{C_{p}\omega }-\frac{C_{s}M^{2}\omega^{3}}{C_{s}^{2}R_{L}^{2}\omega^{2}+{\left(-1+\frac{\omega^{2}}{\omega_{0}^{2}}\right)}^{2}}\right]=0.$$Equation (7) reveals two sub-equations for revealing whole possible solutions of Im (*Z_in_*) as follows:(7a)$$-1+\frac{\omega^{2}}{\omega_{0}^{2}}=0\;$$
and(7b)$$\frac{1}{C_{p}\omega }-\frac{C_{s}M^{2}\omega^{3}}{C_{s}^{2}R_{L}^{2}\omega^{2}+{\left(-1+\frac{\omega^{2}}{\omega_{0}^{2}}\right)}^{2}}=0.$$Sub-equation (7a) yields a solutions of $\omega =\omega_{0}$, which is always real provided that the primary-side compensation condition in Equation (5) is satisfied. Sub-equation (7b) can be simplified as(7c)$$C_{s}^{2}R_{L}^{2}\omega^{2}+{\left(-1+\frac{\omega^{2}}{\omega_{0}^{2}}\right)}^{2}=C_{s}C_{p}M^{2}\omega^{4}.$$Introducing the dimensionless variable $u=\omega^{2}/\omega_{0}^{2}$ and the compact parameters $\beta =C_{s}R_{L}\omega_{0}$, $\kappa ≜\omega_{0}^{2}M\sqrt{C_{s}C_{p}}$, sub-equation (7d) yields the following equation:(7d)$$(u-1{)}^{2}+\beta^{2}u=\kappa^{2}u^{2}.$$Rearranging (7d) finally gives the following quadratic equation:(7e)$$\left(1-\kappa^{2}\right)u^{2}+\left(\beta^{2}-2\right)u+1=0.$$Sub-equation (7d) yields two real roots when the discriminant is positive (Δ > 0) as follows:(8)$$\Delta ={\left(\beta^{2}-2\right)}^{2}-4\left(1-\kappa^{2}\right)=\beta^{4}-4\beta^{2}+4\kappa^{2}.$$By substituting the aforementioned compact parameters and further manipulating (8), the following bifurcation condition (Equation (9)) is obtained [31]. When satisfied, it ensures two real solutions for $\omega_{L}$ and $\omega_{H}$ in Equation (7).(9)$$\sqrt{C_{P}}M\omega >\sqrt{C_{s}}R_{L}.$$These two real solutions, $\omega_{L}$ and $\omega_{H}$, can be obtained by solving the roots for *u* in the quadratic equation of (7e) as follows:(10)$$u_{L,H}=\frac{2-\beta^{2}\mp \sqrt{\beta^{4}-4\beta^{2}+4\kappa^{2}}}{2\left(1-\kappa^{2}\right)}$$Finally, by substituting the aforementioned compact parameters into (10), then real $\omega_{L}$ and $\omega_{H}$ solutions are obtained as given in Equations (11) and (13) (see Table 1).

Table 1 summarizes the solutions and bifurcation conditions derived throughout Equations (6)–(10). The solution frequency $\omega_{0}$ consistently exists and remains invariant to the mutual inductance M and load resistance $R_{L}$, offering a reliable operating point for sensor interrogation. In contrast, the other two ZPA frequencies only appear under the bifurcation condition ($\sqrt{C_{P}}M\omega >\sqrt{C_{s}}R_{L}$) and are sensitive to coupling and loading variations (see Table 1).

The input impedance, $Z_{in}$, becomes pure real at these ZPA frequencies-commonly referred to as $R_{in}$-can be evaluated by substituting the ZPA frequencies (solutions) into Equation (3), and is provided in Equations (14)–(16). Notably, the impedance values at $\omega_{L}$ and $\omega_{H}$ are identical, resulting in a symmetric impedance profile around $\omega_{0}$ in the frequency domain. These frequencies are ordered in ascending manner, where $\omega_{L}$ corresponds to the lowest, and $\omega_{H}$ to the highest resonance frequency, with $\omega_{0}$ located centrally between them. At $\omega_{0},\;R_{in}$ depends primarily on M and $R_{L}$ (refer to Equation (16)) while the values at $\omega_{L}$ and $\omega_{H}$ are influenced by a broader set of parameters including $C_{s},\;L_{s}$, and $C_{p}$ (refer to Equation (14)).(14)$$Z_{in\;}\left(\omega =\omega_{L}\right)=Z_{in\;}\left(\omega =\omega_{H}\right)=R_{in}=\;\frac{{C_{s}}^{2}M^{2}R_{L}\omega_{0}^{4}}{\sqrt{C_{s}}\omega_{0}X+{C_{s}}^{3/2}\left(-L_{s}\right)\omega_{0}^{3}X+C_{s}\left(C_{p}M^{2}\omega_{0}^{4}-2L_{s}\omega_{0}^{2}\right)+{C_{s}}^{3}L_{s}{R_{L}}^{2}\omega_{0}^{4}+{C_{s}}^{2}\left({L_{s}}^{2}\omega_{0}^{4}-{R_{L}}^{2}\omega_{0}^{2}\right)+1}$$ where(15)$$X=\sqrt{\omega_{0}^{2}\left(4C_{p}M^{2}+{C_{s}}^{3}{R_{L}}^{4}\right)-4C_{s}{R_{L}}^{2}}$$
and(16)$$Z_{in\;}\left(\omega =\omega_{0}\right)=R_{in\;}=\frac{M^{2}\omega_{0}^{2}}{R_{L}}$$

In practical implementations (e.g., passive sensor applications as in this work), identical inductors ($L_{p}=L_{s}=L)$ and hence so capacitors ${(C}_{p}=C_{s}=C$), are often used on both sides, which simplifies the analysis. In such case, the solutions of $\omega_{L},\;\omega_{0}$, and $\omega_{H}$, and bifurcation condition and are simplified to the equations in (17)–(19) and are given in Table 2. It is important to emphasize that once the bifurcation condition $M\omega >R_{L}$ is satisfied (refer to Table 2), the sensor’s load resistance $R_{L}$ is effectively and directly reflected to the input impedance at the split zero-phase angle (ZPA) resonance frequencies, namely $\omega_{L}$ and $\omega_{H}$ (see Equation (20)). This direct reflection enables accurate and passive readout of the sensor resistance through impedance measurement at these specific frequencies. The presence of these split resonance frequencies offers a distinct advantage in resistive sensing applications, such as thermistor-based wireless temperature monitoring.
sensors-25-06316-t002_Table 2Table 2The solutions of ωL,ω0, and ωH, and bifurcation condition for identical coils (inductors).Solutions for ZPA Resonance Frequencies
Bifurcation ConditionsωL=122LC−CRL2−Cs−4M2−4CLRL2+C2RL4(LC)2+C2M2(17)ωL is real if Mω>RL
ω0=1CL(18)ω0 is always realωH=122LC−CRL2+Cs−4M2−4CLRL2+C2RL4(LC)2+C2M2(19)ωL is real if Mω>RL

(20)$$Z_{in\;}\left(\omega =\omega_{L}\right.\mathrm{or}\left.\omega_{H}\right)=R_{in\;}=R_{L}$$(21)$$Z_{in\;}\left(\omega =\omega_{0}\right)=\frac{M^{2}\omega_{0}^{2}}{R_{L}}$$

## 3. Practical Implementation of the SS-Compensated LC-Thermistor Sensor Topology

In the experimental implementation of the LC-thermistor sensor topology, three commercially available passive components were employed to construct the resonant circuit: a planar coil, a thermistor, and a ceramic capacitor. The inductor used in both the primary and secondary resonant tanks is a flat spiral coil (model XKT-L3) with an inductance of 14 µH, an outer diameter of 43 mm, a wire diameter of 1 mm, and a thickness of 2.3 mm. Its symmetrical design ensures consistent magnetic coupling and facilitates direct reflection of the sensing resistance to the input. As the temperature-sensitive element, an NTC thermistor (model NTC10D-9) with a nominal resistance of 10 Ω at 25 °C and a 9 mm disk diameter was integrated into the secondary circuit. The thermistor’s resistance exhibits a negative exponential relationship with temperature, enabling contactless thermal sensing via changes in the system’s input impedance. For resonance tuning, identical leaded safety capacitors were used on both the primary and secondary sides: Murata DE1B3KX331KA4BP01F, 330 pF, K tolerance (±10%), dielectric type B (JIS), X1/Y1 class, rated 300 Vac (r.m.s.). According to the datasheet, the B (JIS) characteristic specifies a capacitance change of ±10% over −25 °C to 85 °C, with an operating temperature range of −40 °C to 125 °C.

The nominal values of the aforementioned components correspond to their ideal low-frequency specifications. However, at radio frequencies (RF), these components exhibit significant parasitic behaviors—such as stray capacitance, lead inductance, skin and proximity effects, and dielectric losses—that can substantially alter their performance. As a result, accurate characterization of each component at the intended operating frequency is critical for reliable circuit design and simulation. Unfortunately, RF models or manufacturer-provided S-parameter data for the selected components were not readily available. To address this limitation, each component (coil, thermistor, and capacitor) was RF-modeled as discussed in following subsection.

### 3.1. RF-Modeling of Utilized Components

The components are characterized as a one-port microwave network using a ROHDE & SCHWARZ ZNLE6 vector network analyzer over the 1.5 MHz to 5 MHz frequency range. Prior to measurement, a precise open-short-load (OSL) calibration was performed at the end of the measurement cable to ensure high accuracy. Figure 2 illustrates the measurement setup and shows a component connected to the network analyzer during testing.

The one port frequency-dependent S-parameter, *S*_11_(*ω*), was measured for each component during aforementioned one-measurement and is converted to impedance via(22)$$Z\left(\omega \right)=Z_{0}\frac{1+S_{11}\left(\omega \right)}{1-S_{11}\left(\omega \right)}\;\;\left(Z_{0}=50\;\Omega \right).$$Then, the measured impedance is fitted to two or three element lumped element for achieving best model-measurement agreement.

The XKT-L3 coil was first characterized through direct measurement and subsequently modeled using a three-element equivalent inductor model as given in Figure 3a. This model consists of a series combination of an inductor and a resistor, both of which are in parallel with a capacitor, effectively capturing the coil’s frequency-dependent behavior at RF. This three-element model is a common RF model for inductors operating at high frequencies. The resistor represents the DC and AC resistance of the inductor winding, while the capacitor accounts for the cumulative capacitive effect of the inter-winding capacitance. The inductance value in the model is directly obtained from LCR meter measurements. The capacitor is then manually tuned to achieve the best agreement between the modeled and measured reactance. Finally, the resistor value is fitted to a function using curve fitting tools, followed by minor manual adjustments. The resulting model-to-measurement fitting curves—depicting both the real and imaginary components of the input impedance—are illustrated in Figure 3b and Figure 3c, respectively. The resistor in the model is frequency-dependent, primarily due to the skin and proximity effects that dominate at radio frequencies, and its behavior is approximated by a fitted mathematical expression, as shown in Equation (23).(23)$$R=3.013\cdot 10^{-12}\cdot f^{1.905}$$

Subsequently, the capacitor was characterized using one-port S-parameter measurements obtained via the calibrated network analyzer. Analysis of the measurement results revealed that, in addition to its expected capacitive reactance, the component exhibits a frequency-dependent resistive behavior—likely due to dielectric and electrode losses increasing with frequency. To account for this, a two-element model comprising a series connection of a capacitor and a frequency-dependent resistor, of which frequency-dependency is expressed as given in Equation (24), was adopted to accurately represent the RF behavior of the compensation capacitors used on both the primary and secondary sides of the circuit. The modeling results are compared with experimental data in Figure 4. As illustrated in Figure 4b, a minor discontinuity is observed near 3 MHz that is not fully captured by the model; however, the overall agreement between the measured and simulated impedance responses is notably strong.(24)$$R=\frac{2.28\times 10^{7}}{f}+1.40\times 10^{-6}\cdot f-6.51$$

Finally, the thermistor is modeled using a simple two-element configuration consisting of a frequency-dependent series-connected resistor (*R*(*f*)) (see Equation (25)) and a frequency-dependent positive reactance (*Z*(*f*)) (see Equation (26)). The presence of inductive reactance is attributed to the physical structure and lead geometry of the thermistor, which introduces parasitic inductive effects under RF excitation. The modeling results are compared with the measured impedance data in Figure 5, where both the real and imaginary parts of the input impedance are shown across the frequency range of interest. As seen in Figure 5a,b, the model closely matches the measurement data, capturing the resistive impedance variation as well as the positive slope of the imaginary component. This minimal yet effective model proves sufficient for accurately representing the thermistor’s behavior within the operating frequency range of the sensor system.(25)$$R(f)=\frac{9.88\times 10^{5}}{f}+1.59\times 10^{-7}\cdot f+9.07$$
(26)$$Z(f)=j\cdot \left(\frac{-4\times 10^{5}}{f}+1.07\times 10^{-7}\cdot f+0.3)\right.$$

To establish the intrinsic temperature dependence of the employed NTC element, we measured the thermistor resistance using a benchtop LCR meter over 25–40 °C. Figure 6 below shows a graph of measured NTC resistance vs. temperature. The resistance decreases monotonically from approximately 10–11 Ω at 25 °C to near 6 Ω at 40 °C, consistent with NTC behavior.

### 3.2. LC-Thermistor Sensor Implementation

The LC-thermistor topology shown in Figure 1 was experimentally implemented in a laboratory setting. The experiments were conducted inside a cubical chamber constructed from sintered glass, designed to provide a controlled high-temperature environment. Heated airflow from a heat gun was introduced through an opening at the top of the chamber, while a thermometer placed inside enabled continuous temperature monitoring during testing.

Using this chamber, identical XKT-L3 coils and identical series-connected compensation capacitors were employed on both the primary and secondary sides, enabling direct reflection of the thermistor resistance from the sensor side to the input—forming the core operational principle of the study. The secondary side was placed inside the chamber while the primary side was outside for wireless measurement of the in-chamber temperature. The coils were precisely aligned face-to-face on opposite sides of the glass wall, separated by its 10 mm thickness. This separation resulted in a mutual inductance *M* of 3.75 µH, as determined from coupling measurements. Figure 7 shows a photograph of the experimental setup in laboratory.

The system is configured to operate at a fundamental resonance frequency of 2.27 MHz, chosen based on the availability of off-the-shelf component values, meeting the bifurcation condition, $M\omega >R_{L}$, (see Equations (17) and (19)) and the lower frequency limit of the network analyzer used for characterization. The updated schematic, incorporating the RF models of the components described in detail above, is presented in Figure 8. At room temperature—when no heating is applied by the heat gun—the temperature inside the chamber was recorded as 25.8 °C. Under these conditions, the measured input impedance and the simulated impedance obtained using the model in Figure 8 are compared in Figure 9. As shown, the model and measurement results exhibit excellent agreement, underscoring the critical importance of accounting for the RF characteristics of the components used in the system. It should be noted that mechanical perturbations (microphonics or vibration) could in principle modulate $C_{p/s}$ or the coupling k, but in our bench-top tests the components were rigidly mounted and measurements were taken under steady-state conditions; no modulation of the split frequencies was observed within the VNA resolution.

As shown in Figure 9, the split resonance frequencies *f_L_* and *f_H_*, along with the system’s main resonance frequency *f*_0_, are indicated by dashed lines in both plots. These frequencies correspond to the points where the imaginary part of the input impedance crosses zero (see Figure 9b). The measured values for *f_L_*, *f*_0_, and *f_H_* are 2.045 MHz, 2.272 MHz, and 2.637 MHz, respectively.

The final stage of the laboratory experiment involved increasing the temperature inside the glass chamber using a heat gun, as illustrated in Figure 7. The temperature was gradually raised to approximately 40 °C while being continuously monitored with a thermometer. Figure 10 shows the wirelessly measured input resistance ($R_{in}$) outside the chamber as a function of the in-chamber temperature at the split resonance frequencies $f_{L}$ and $f_{H}$. At 25.8 °C, the measured $R_{\mathrm{in}}$ values at both $f_{L}$ and $f_{H}$ are identical, which agrees with the theoretical expectation given in Equation (20). It is important to note that the measured $R_{\mathrm{in}}$ values include the cumulative effect of RF parasitic resistances originating from the compensation capacitors and coils. As a result, the $R_{in}$ at room temperature starts at approximately ~36–37 Ω, despite the nominal thermistor load resistance being 10 Ω. This behavior is also evident in the model-measurement comparison plots in Figure 9a, where the resistance values at $f_{L}$ and $f_{H}$ are clearly offset from the ideal.

As the temperature increases, the $R_{in}$ values at $f_{L}$ and $f_{H}$ diverge, creating an asymmetry around the main resonance frequency $f_{0}$. This phenomenon is attributed to the secondary-side compensation capacitor, which is located inside the heated chamber and therefore directly exposed to temperature variations. With rising temperature, the capacitor’s value shifts, introducing a perturbation [24] that alters the resonance condition of the coupled system and breaks its symmetry. Consequently, this shift causes the $R_{in}$ to decrease at one split frequency while increasing by a similar amount at the other. When the secondary compensation capacitor experiences a small drift, $C_{s}\to C_{s}(1+\delta )$ with |δ|≪1, the input-resistance readings at the split frequencies respond with opposite sign sensitivities, i.e., $R_{in}\left(f_{L}\right)=R_{0}+\alpha \delta +O\left(\delta^{2}\right)$ and $R_{in}\left(f_{H}\right)=R_{0}-\alpha \delta +O\left(\delta^{2}\right)$, where $R_{0}$ is the input resistance at near room temperature (25 °C). Consequently, their average(27)$$R_{avg}=\frac{1}{2}\left(R_{in}\left(f_{L}\right)+R_{in}\left(f_{H}\right)\right)=R_{0}+O\left(\delta^{2}\right)$$
is first-order insensitive to such drifts, explaining the near-constancy of $R_{\mathrm{avg}}$. For example, with $L_{p}=L_{s}=14\;\mu H,\;C_{p}=C_{s}=330\;pF,\;R_{L}=10\;\Omega$, and M=3.75 μH, we evaluate the splitpoint resistances from the closed-form ZPA solutions and $Z_{in}$ expression. A ±1% change in $C_{s}$ yields the following Table (Table 3).

As a result, the average value of measured $R_{in}$ values at the split frequencies as calculated below(28)$$R_{in,avg}=\frac{R_{in}\left(f=f_{H}\right)+R_{in}\left(f=f_{L}\right)}{2}$$
remains close to the value expected in a perfectly symmetric system, as shown in Figure 10. This makes $R_{in,avg}$ also a preferred sensing parameter in scenarios where the reactive components of the circuit are subject to environmental influences such as temperature drift.

To check repeatability, the temperature sweep was performed two additional times at the same set-points (total n=3). Figure 11 overlays the average split-point resistance, $R_{avg}=\frac{1}{2}\left[R_{in}\left(f_{L}\right)+R_{in}\left(f_{H}\right)\right]$, for the three independent runs. The traces closely coincide over 25–40 °C and follow the same monotonic trend.

Another wirelessly measurable parameter is the input resistance at the main resonance frequency, $R_{\mathrm{in}}(f=f_{0})$, whose relationship to the load thermistor resistance is given in Equation (21). As predicted by this equation, $R_{in}\left(f=f_{0}\right)$ increases as the thermistor resistance decreases with rising temperature, exhibiting an inverse proportionality between the two. This trend is clearly demonstrated in Figure 12, where a pronounced change in $R_{\mathrm{in}}\left(f=f_{0}\right)$ is observed over the tested temperature range. More importantly, the $f_{0}$ is independent of load resistance, making it suitable for reader circuits with single tone signal source.

An important implication of this result is the flexibility it offers in sensor readout design. While $R_{\mathrm{in}}$ at $f_{0}$ provides a direct and simplified method of temperature measurement, the coupled resonator system inherently allows similar resistance measurements to be made at the split resonance frequencies $f_{L}$ and $f_{H}$. Depending on the intended application and readout circuitry, the designer can choose to measure at any of these three resonance points. For example, selecting $f_{0}$ may be advantageous for single-frequency interrogation systems where circuit simplicity is critical, whereas monitoring $f_{L}$ or $f_{H}$ could provide additional sensing robustness or redundancy, particularly in environments where resonance frequency drift or asymmetry might occur. This versatility enhances the practicality of the proposed SS-compensated LCthermistor topology for a wide range of passive wireless sensing applications.

### 3.3. Comparison with Various Sensors

Table 4 compares recent passive LC temperature sensors [12,36,37,38] with the present SS impedance-mirroring design. Prior LC devices read temperature via a resonance shift and therefore need a swept/PLL (VNA-like) reader, so the readout depends on Q, parasitics, and coupling. Our sensor instead operates at the split ZPA frequencies $f_{L},f_{H}$, where the thermistor resistance is mirrored at the input; a single/two-tone narrowband reader suffices. Thus, the proposed approach offers a simpler reader and an IC-free, battery-free tag, while differences in sensing material, topology, and packaging across [12,36,37,38] explain the spread in reported performance. The current 25–40 °C validation serves as a proof-of-concept; wider range and distance are engineering extensions rather than fundamental limits.

## 4. Discussion

The proposed SS-compensated LC-thermistor topology demonstrates a distinct advantage over conventional passive wireless sensing configurations by directly reflecting the thermistor resistance at the split resonance frequencies of the strongly coupled system. This feature, confirmed both analytically and experimentally, enables accurate, battery-free, and contactless monitoring of temperature-dependent resistance. The modeling results presented in Section 2, particularly Equations (11)–(21), show that under the bifurcation condition, two additional zero-phase-angle (ZPA) frequencies emerge symmetrically around the original resonance. This symmetry and the predictable impedance values at the split points make the topology particularly robust for resistive sensing applications.

The RF modeling of individual components (Figure 3, Figure 4 and Figure 5) played a crucial role in ensuring the accuracy of the system’s predicted performance. The extracted parameters for the inductors, capacitors, and thermistors confirmed frequency-dependent behavior that, when incorporated into the circuit model, provided excellent agreement between simulation and measurement as seen in Figure 9. The accurate representation of parasitic effects, such as series resistance and electrode-induced inductance, was essential to matching experimental outcomes with theoretical predictions.

One of the key observations from the temperature variation experiments is that secondary side compensation capacitor also varies as NTC thermistor’s resistance decreases with increasing temperature, leading to a perturbation in the coupled system that shifts the split frequencies unequally. This phenomenon, while small, suggests that environmental factors and component tolerances can induce asymmetries not accounted for in idealized models—a point also noted in earlier experimental studies of bifurcated resonance systems [24].

From an application standpoint, the ability to track sensor resistance simply by monitoring the input resistance at the split frequencies opens up promising opportunities in wireless condition monitoring. For example, the topology could be adapted for structural health monitoring by substituting the thermistor with a strain-dependent resistive sensor, or for chemical sensing using resistive films whose impedance varies with gas concentration. In addition, because the impedance mirroring effect is relatively insensitive to absolute coupling variations (as long as the bifurcation condition is satisfied), the system can tolerate moderate misalignment between reader and sensor—a significant advantage over traditional single-resonator systems.

External RF fields can be coupled into the reader/tag coils and appear as a small ripple or offset in the measured input impedance. In our benchtop tests, conducted in a low-noise lab environment with short leads, we did not observe EMI-induced variations. Practically, susceptibility is reduced by the narrowband which further suppresses out-of-band or common-mode interference.

Future research directions could focus on expanding the operational frequency range, miniaturizing the resonators for integration into compact sensor tags, and developing multi-parameter sensing schemes by combining resistive sensing with capacitive or inductive modalities in a hybrid topology. Moreover, while the current work validates the concept under laboratory conditions, field testing in realistic application environments (e.g., rotating machinery, embedded biomedical implants, or submerged monitoring stations) would further demonstrate the robustness and practicality of the approach.

## 5. Conclusions

This work has presented the design, modeling, and experimental validation of a passive wireless temperature sensing system based on an SS-compensated LC-thermistor topology. The system leverages the impedance mirroring property of the SS configuration to directly reflect the thermistor resistance at the split resonance frequencies, enabling accurate, battery-free, and contactless measurement. Detailed RF modeling of the constituent components ensured that the analytical predictions closely matched the measured system response. Experimental results confirmed that the topology maintains strong agreement between theoretical and practical behavior. Beyond temperature monitoring, the same principle could be extended to a wide range of resistive sensors, opening avenues for structural, environmental, and biomedical applications. The results suggest that the SS-compensated LC topology is a strong candidate for future low-power, high-reliability wireless sensing platforms.

In our proof-of-concept prototype, the demonstrated operating temperature span was 25–40 °C (Figure 10), over which the thermistor exhibited an average slope of |dR/dT|≈0.3 Ω/°C (Figure 6). Using this measured slope and a conservative impedance repeatability of δR≈0.3 Ω at the split points. The proposed LC-thermistor sensor can also be interrogated without a VNA. A low-cost RF signal generator combined with a small series shunt resistor allows simultaneous measurement of sensor voltage and current. From these signals, the input impedance—and hence the resonance shift—can be extracted, providing a portable and cost-effective readout solution.

In terms of application, the SS-based, IC-free tag with a sweep-free (narrowband) reader is well suited for in situ temperature tracking in sealed or thermally isolated spaces-e.g., across non-metallic walls or containers where a few-millimeter to centimeter inductive gap is available. The same principle extends beyond temperature: by placing an appropriate resistive transducer on the secondary side, the system can sense other quantities such as mechanical strain (strain gauge embedded in concrete), light intensity (LDR in a closed chamber), or humidity (humistor in harsh, cable-inaccessible locations).

Current limitations stem primarily from inductive coupling and from the resistance range required to satisfy the bifurcation condition. Practical mitigations include larger or ferrite-backed coils, optimized matching and operating frequency, litz conductors to increase Q, and magnetic shielding or spacers near metal structures; selecting an appropriate thermistor/package also facilitates operation over wider temperature ranges. Future work will address extended span (e.g., −10–100 °C), long-term stability and hysteresis via multi-cycle heating/cooling, multi-sensor readout, and a compact single- or two-tone reader implementation.

## Figures and Tables

**Figure 1 sensors-25-06316-f001:**
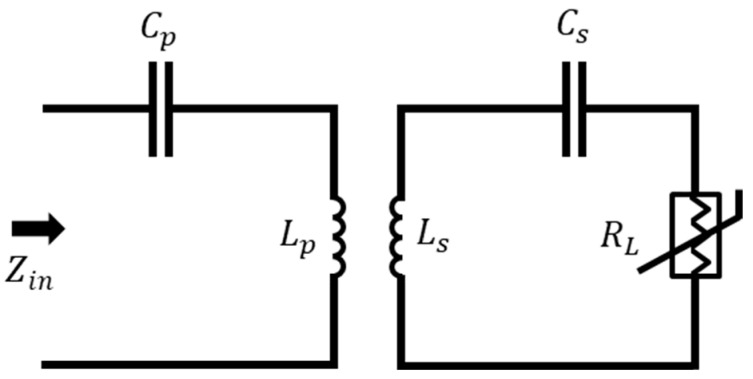
Circuit model for the SS-compensated LC-Thermistor model.

**Figure 2 sensors-25-06316-f002:**
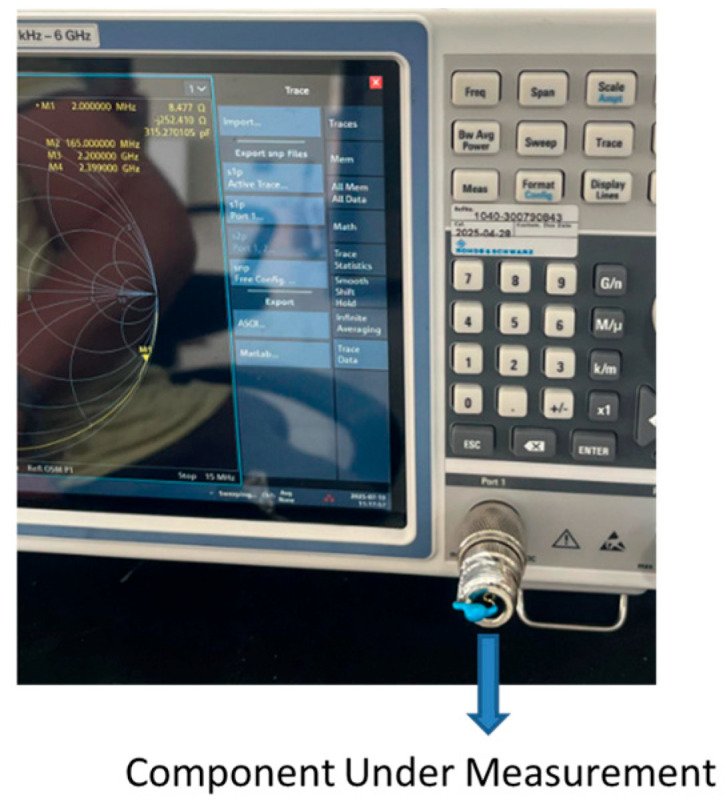
Measurement setup for component testing.

**Figure 3 sensors-25-06316-f003:**
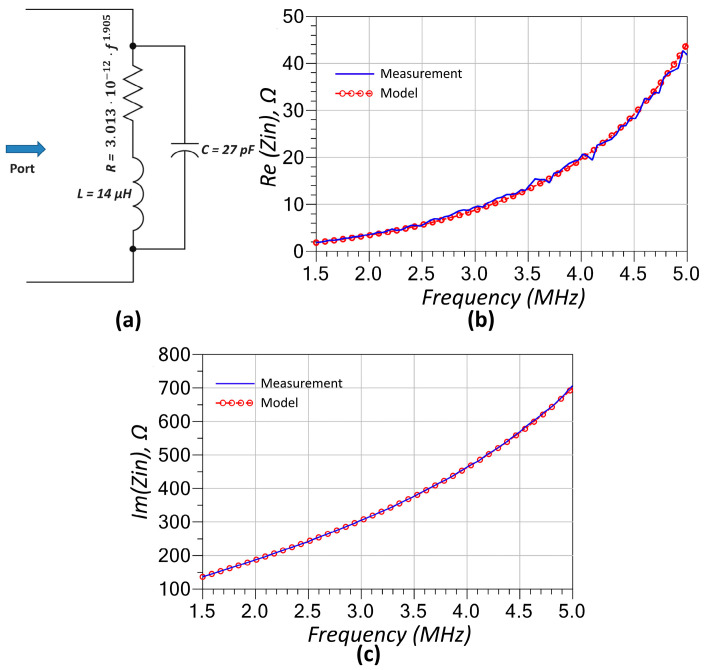
(**a**) Equivalent RF model of the inductor based on a three-element representation. Comparison of the (**b**) real part and (**c**) the imaginary part of the input impedance obtained from the model and one-port component measurements, demonstrating strong agreement across the frequency range.

**Figure 4 sensors-25-06316-f004:**
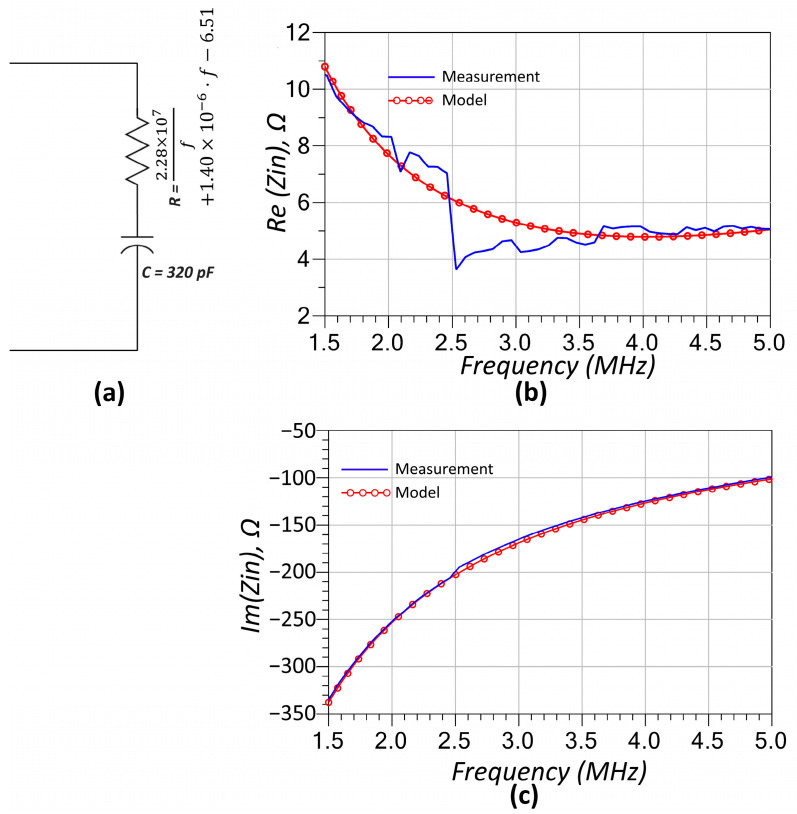
(**a**) Equivalent RF model of the capacitor based on a two-element representation. (**b**) Comparison of the real part and (**c**) the imaginary part of the input impedance obtained from the model and one-port component measurements, demonstrating strong agreement across the frequency range.

**Figure 5 sensors-25-06316-f005:**
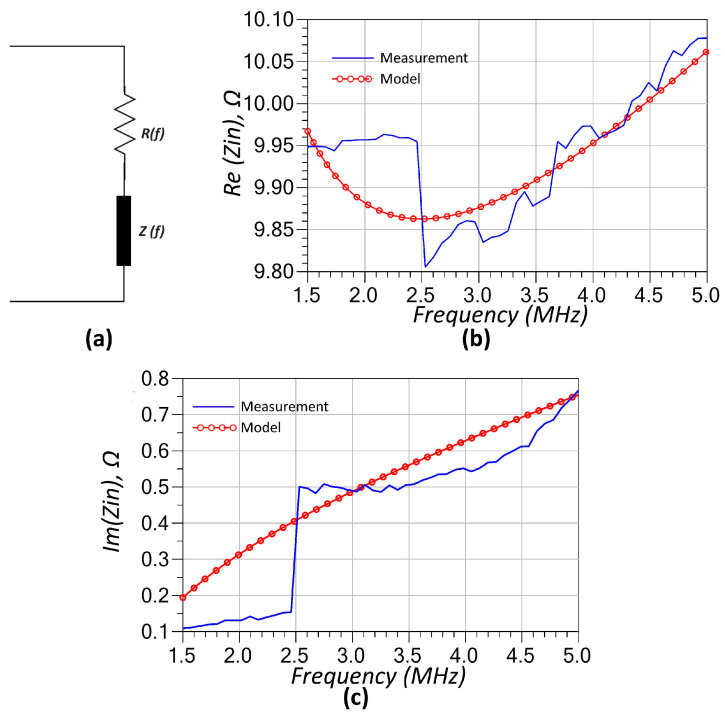
(**a**) Equivalent RF model of the thermistor based on a two-element representation. (**b**) Comparison of the real part and (**c**) the imaginary part of the input impedance obtained from the model and one-port component measurements, demonstrating good agreement across the frequency range. The functions *R*(*f*) and *Z*(*f*) are given in Equations (25) and (26).

**Figure 6 sensors-25-06316-f006:**
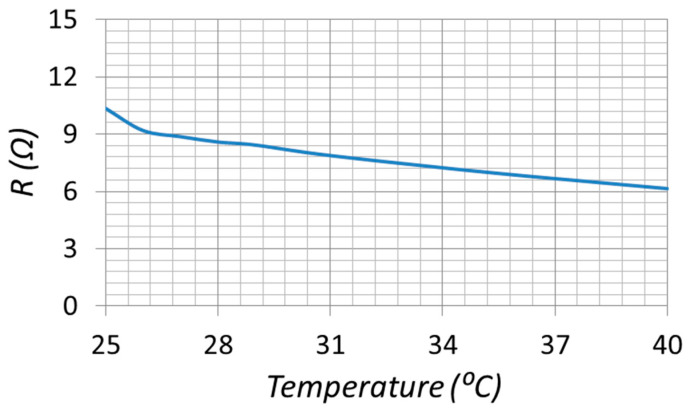
Measured NTC resistance vs. temperature.

**Figure 7 sensors-25-06316-f007:**
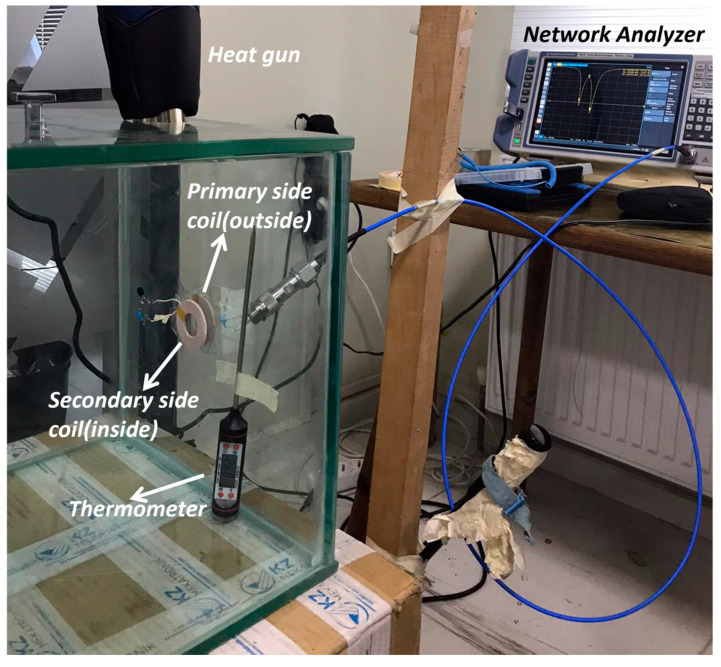
Photograph of the experimental setup for SS-compensated LC-Thermistor sensor. The secondary coil (inside the chamber) and the primary coil (outside) are aligned face-to-face across the glass wall; the glass thickness is 10 mm, which sets the coil-to-coil spacing and yields a measured mutual inductance of approximately *M* = 3.75 μH.

**Figure 8 sensors-25-06316-f008:**
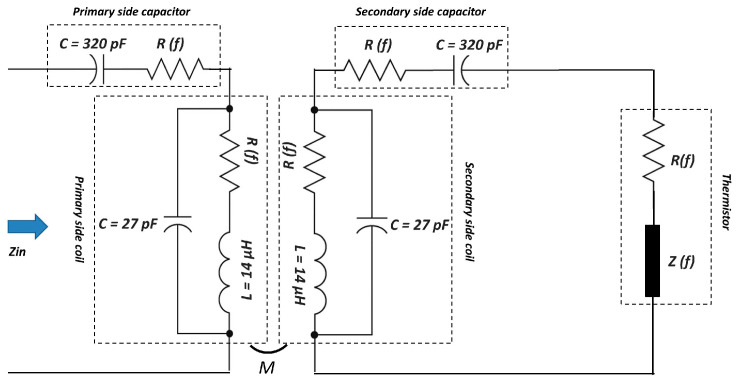
Schematic of the SS-compensated LC-Thermistor with RF modeled components.

**Figure 9 sensors-25-06316-f009:**
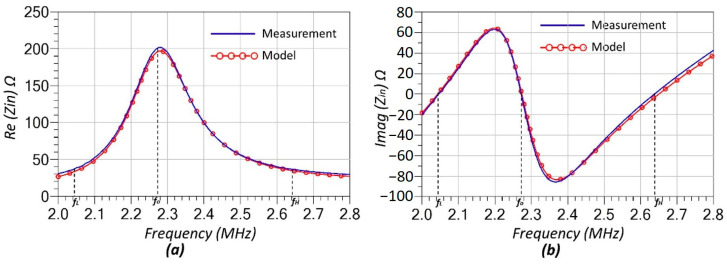
Comparison of measured and simulated input impedance of SS-Compensated LC-Thermistor. (**a**) Real part of the input impedance (*R_in_*) and (**b**) imaginary part of the input impedance (*X_in_*) at room temperature (25.8 °C).

**Figure 10 sensors-25-06316-f010:**
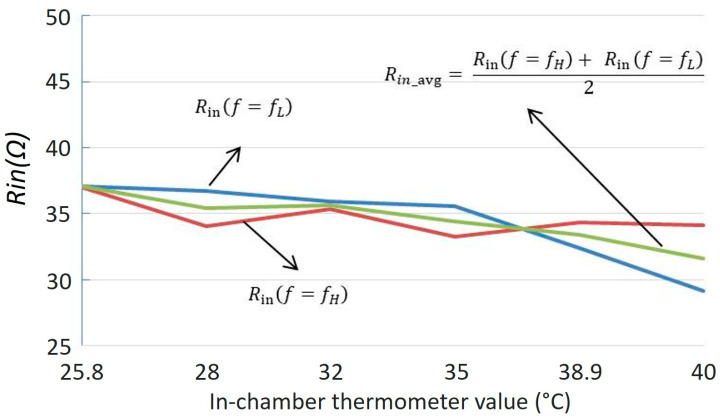
Measured input resistance from input of the SS-Compensated LC-Thermistor at split resonance frequencies, *f_L_* and *f_H_*. The average of the input resistances at these frequencies are also given in the graph.

**Figure 11 sensors-25-06316-f011:**
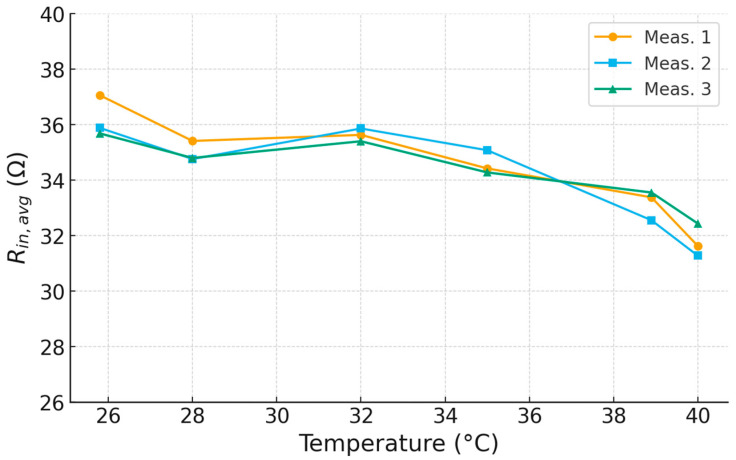
Overlay of three independent runs of $R_{\mathrm{avg}\;}=\frac{1}{2}\left[R_{in}\left(f_{L}\right)+R_{in}\left(f_{H}\right)\right]$ versus temperature.

**Figure 12 sensors-25-06316-f012:**
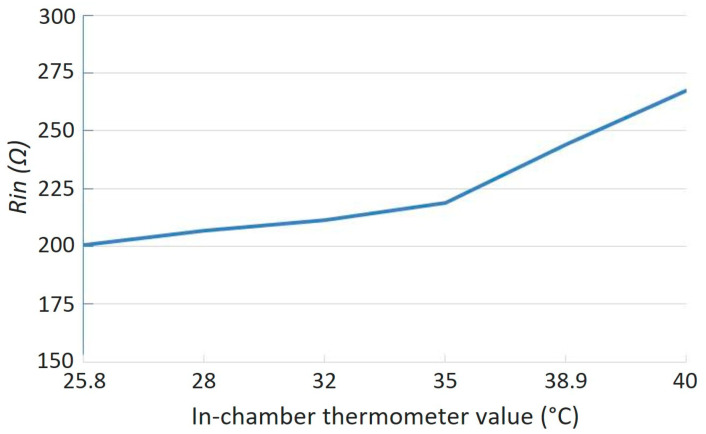
Measured input resistance from input of the SS-Compensated LC-Thermistor at main resonance frequency *f*_0_.

**Table 1 sensors-25-06316-t001:** The solutions of $\;\omega_{L},\omega_{0}$, and $\omega_{H}$, and bifurcation condition.

Solutions for ZPA Resonance Frequencies		Bifurcation Conditions
$\omega_{L}=\frac{1}{\sqrt{2}}\sqrt{\frac{-2\omega_{0}^{2}+{C_{s}}^{2}{R_{L}}^{2}\omega_{0}^{4}-\sqrt{C_{s}}\omega_{0}^{3}\sqrt{-4C_{s}{R_{L}}^{2}+4C_{p}M^{2}\omega_{0}^{2}+{C_{s}}^{3}{R_{L}}^{2}\omega_{0}^{2}}}{-1+C_{p}C_{s}M^{2}\omega_{0}^{4}}}$	(11)	$\omega_{L}$ is real if $\sqrt{C_{p}}M\omega >\sqrt{C_{s}}R_{L}$
$\omega_{0}=\frac{1}{\sqrt{C_{s}L_{s}}}=\frac{1}{\sqrt{C_{p}L_{p}}}$	(12)	$\omega_{0}$ is always real
$\omega_{H}=\frac{1}{\sqrt{2}}\sqrt{\frac{-2\omega_{0}^{2}+{C_{s}}^{2}{R_{L}}^{2}\omega_{0}^{4}+\sqrt{C_{s}}\omega_{0}^{3}\sqrt{-4C_{s}{R_{L}}^{2}+4C_{p}M^{2}\omega_{0}^{2}+{C_{s}}^{3}{R_{L}}^{2}\omega_{0}^{2}}}{-1+C_{p}C_{s}M^{2}\omega_{0}^{4}}}$	(13)	$\omega_{H}$ is real if $\sqrt{C_{p}}M\omega >\sqrt{C_{s}}R_{L}$

**Table 3 sensors-25-06316-t003:** Effect of ±1% drift in the secondary compensation capacitor $C_{s}$ on the input resistance at the split frequencies $R_{in}\left(f_{L}\right)$ and $R_{in}\left(f_{H}\right)$, and on the average readout $R_{\mathrm{avg}}$; the opposite-signed changes at $f_{L}$ and $f_{H}$ illustrate first-order cancelation in $R_{\mathrm{avg}}$.

Case	$C_{s}$	$R_{\mathrm{in}\;}\left(f_{L}\right)$	$R_{\mathrm{in}\;}\left(f_{H}\right)$	$R_{\mathrm{avg}}$
Nominal	330 pF	10.00 Ω	10.00 Ω	10.00 Ω
$C_{s}↓1\%$	$0.99\times C_{s}$	9.53 Ω	10.28 Ω	9.91 Ω
$C_{s}↑1\%$	$1.01\times C_{s}$	10.49 Ω	9.73 Ω	10.11 Ω

**Table 4 sensors-25-06316-t004:** Side-by-side comparison of various representative temperature sensors and this work.

Ref	Topology/Readout	Sensing Element	Range (°C)	Sensitivity	Reader Complexity
[36]	Near field coupling of RF antenna link.	PCB patch antenna	−40 to 125	347.45 kHz/°C	Sweep/PLL or VNA
[37]	LC passive, dual-layer inductor (L)	PEG dielectric between dual inductors (biocompatible/biodegradable)	36 to 40	4.11 MHz/°C	Sweep/PLL or VNA;
[38]	Single-LC, frequency-shift	Graphene-oxide capacitive film	−40 to 0 and 10 to 60	59.3 kHz/°C (−40–0), 46.1 Hz/°C (10–60)	Sweep/PLL or VNA;
[12]	Passive LC (dual-sensitive element), bearing apps	Dual sensing elements (both C and L)	−40 to 140	**No ferrite:** −40 °C–40 °C: −14.91 kHz/°C 40 °C–100 °C: +37.59 kHz/°C ------------------ **With ferrite:** −40 °C–40 °C: −17.12 kHz/°C 40–100 °C: +139.32 kHz/°C 100–140 °C: +72.23 kHz/°C	Sweep/PLL or VNA;
This work	Impedance mirroring	NTC thermistor	25–40	0.30 Ω/°C (from $R_{\mathrm{avg}}$)	Single tone or two-tone source or VNA

## Data Availability

The raw data supporting the conclusions of this article will be made available by the authors on request.

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
