# Peer review of "Implementation of an SS-Compensated LC-Thermistor Topology for Passive Wireless Temperature Sensing"

_sensors, 2025, doi:10.3390/s25206316_

Round 1
Reviewer 1 Report
Comments and Suggestions for Authors
1. Introduction lacks a clear statement of novelty.
While the background on passive wireless sensors and PT-symmetry is adequately covered, the specific contribution of this work relative to existing SS-compensated topologies is not sufficiently highlighted. Please explicitly state what distinguishes this implementation from prior art.
2. Mathematical derivations need further clarification.
The equations in Section 2 (especially Eqs. 11–13 and Table 1) are complex and lack step-by-step derivations. Providing intermediate steps or a supplementary document with detailed derivations would improve reproducibility and reader comprehension.
3. RF modeling methodology requires more detail.
The process of extracting frequency-dependent parameters (e.g., Eqs. 19–22) is not fully explained. Please include information on the fitting procedure, error metrics, and validation methods to ensure the models are robust and reproducible.
4. Experimental validation is limited in temperature range.
The temperature test only goes up to 40°C, which may not be sufficient for many practical applications (e.g., industrial or biomedical). Extending the range and including more data points would strengthen the conclusions.
5. Asymmetry in Rin at split frequencies is not fully analyzed.
The observed asymmetry due to temperature drift of the compensation capacitor is an important finding, but its impact on sensor accuracy and calibration is not thoroughly discussed. A deeper analysis or compensation strategy should be proposed.
6. Comparison with existing sensing methods is missing.
The paper would benefit from a comparative analysis with other passive wireless temperature sensing techniques (e.g., frequency-shift-based LC sensors) to highlight the advantages of the SS-compensated approach in terms of sensitivity, stability, or simplicity.
7. Language and formatting issues need attention.
There are several typos (e.g., “MMM” instead of “M” on page 10), inconsistent figure references, and incomplete author information. The manuscript should be carefully proofread and formatted according to journal guidelines.
Reviewer 2 Report
Comments and Suggestions for Authors
A passive wireless temperature sensor based on an SS-compensated LC-thermistor topology is proposed in this article. However, the article lacks in-depth analysis, making it difficult to reflect theoretical value and innovation.
- The manuscript format must be modified, such as the Abstract being only one paragraph.
- In the Abstract, the author needs to highlight the research advantages by using experimental results.
- The Introduction must be rewritten to comprehensively present the current research. I don't think it's necessary to cite 17 papers in the first sentence.
- Figures 1 and 2 are too rough and need to be replaced.
- The formulas are not standardized and lack analysis and explanation of the theory.
- In my opinion, there is a significant gap between the model results and the measurement results in Figures 5 and 6.
- The Figure number is incorrect.
- Lack of repeated experiments, error values should be indicated in Figures 9 and 10.
Reviewer 3 Report
Comments and Suggestions for Authors
This manuscript proposes a SS-compensated LC topology for a passive wireless temperature sensor, employing an NTC thermistor as the temperature-sensitive element. The temperature-dependent impedance variation is wirelessly transferred to the reader side via magnetic coupling. The experimental conditions are relatively simple, and the work lacks quantitative comparison with other LC topologies and validation under diverse environments. My specific comments are as follows:
- It is recommended that the authors perform a direct resistance–temperature (R–T) measurement for the employed NTC thermistor within the same temperature range as used in the experiments. A calibration curve and corresponding fitting formula should be provided to clarify the intrinsic temperature characteristics of the sensing element.
- The current heating method using a hot air gun and glass chamber introduces significant thermal inertia and airflow disturbances, which may compromise temperature control accuracy and measurement repeatability. A more stable and precise temperature control setup is recommended.
- Although the authors claim advantages of the SS-compensated LC topology over traditional designs, no systematic quantitative comparison is provided. It is suggested to benchmark against common series, parallel, and other compensated LC topologies in terms of sensitivity (Hz/°C or Ω/°C), readout distance, and interference immunity.
- According to the figures, the hot air gun experiment only covers approximately 25–40 ° Without testing under more complex or extreme operating conditions, the claim that the design is suitable for “industrial and biomedical” applications is not sufficiently substantiated.
- The presented plots only show single measurement curves, without error bars or standard deviation analysis from repeated measurements. It is recommended to include repeated measurements and statistical error analysis to evaluate the stability and repeatability of the sensing performance.
Reviewer 4 Report
Comments and Suggestions for Authors
Dear Authors,
The presented research is dedicated to an important problem of wireless battery-free temperature sensor design. The paper proposes an original design of the wireless sensor based on contactless impedance monitoring of an LCR-circuit. The modeling of the sensor frequency response is supported by practical measurements. The working prototype of the sensor was designed and tested. The subject covered in the presented research would be interesting for Sensors journal readers and may be recommended for publication after minor revisions. Below are my specific comments:
Comment 1: Authors should explicitly specify the achieved temperature range and expected precision of the proposed sensor in the Conclusion.
Comment 2: The capacitor type is specified around the line 217. It would be better to explicitly specify the dielectric type (I assume NP0) and tolerance.
Comment 3: The influence of the mechanical properties of the spiral coil and capacitors on sensor performance is not estimated. For example vibration or microphonic effect may affect the resonant frequency and magnetic coupling operation. Some short note could be added.
Comment 4: The influence of electromagnetic interference (EMI) on the sensor is not considered. A short note could be added in Discussion section.
Comment 5: A short note how to design a readout circuit for the proposed sensor without using expensive VNA equipment could be added in conclusion.
Comment 6: The test setup photo (Fig. 6) contains black text on black background fragments. Using a contrast color would improve readability.
Reviewer 5 Report
Comments and Suggestions for Authors
This paper presents the analysis, design, implementation and characterization of a series-series LC thermistor topology for passive sensing the temperature parameter.
In general, the concept is fine but there are several concerns of the work. They are given as follows:
- What is the requirement of passive sensor sensitivity in order to allow it to operate in the proposed topology?
- What type of temperature sensors that cannot be applied in this application in view of different sensitivities?
- Why is it necessary the use of thermistor because it is fragile in terms of the environmental factors such as chemical, moisture and mechanical stress etc..?
- The long term stability of thermistor is also another issue for temperature sensing.
- The use of negative TC thermistor will suffer from nonlinearity in sensing as well as different sensitivities in context of different temperature ranges.
- Calibration of components for practical use seems including too many steps or characterization. This implies that it will increase the test cost or lack of practicality for production.
- Component mismatch always exists in realistic implementation. How do you address this problem?
- There is no performance comparison with other works to support the arguments from the authors' proposed work.
Round 2
Reviewer 1 Report
Comments and Suggestions for Authors
1. The innovation and contribution need to be more clearly defined.
Opinion: The paper repeatedly mentions "novelty" and "high performance" in the introduction and abstract sections. However, it fails to clearly and quantitatively explain what the core innovation of this research is compared to existing technologies (such as traditional LC resonant temperature sensors, RFID-based sensors, etc.). Is it higher sensitivity, lower cost, simpler structure, or better anti-interference ability? 。 This leads to the contribution of the paper appearing ambiguous.
Suggestion: Add a brief comparison table or paragraph in the introduction section, clearly listing the advantages and disadvantages of several mainstream passive wireless temperature sensing technologies. On this basis, precisely position the innovation of this research and the specific technical bottlenecks it aims to solve.
2. The completeness of the experimental design and verification needs to be enhanced.
Opinion: The experimental section was only tested within a relatively narrow temperature range (25°C - 45°C), which although covering some biomedical application scenarios, is insufficient to prove the reliability and linearity of the sensor in a wider temperature range (such as -10°C to 100°C required for industrial monitoring). Additionally, there is a lack of test data on the sensor's repeatability, long-term stability, and hysteresis effect.
It is suggested that the calibration curve within a wider temperature range be supplemented. At least one more set of repeatability experiments (such as conducting three heating and cooling cycles on the same sensor) and short-term stability experiments (such as continuously monitoring the readings for several hours at a constant temperature) be added to enhance the credibility of the results.
3. The depth of comparative analysis with related work is insufficient.
Suggestion: In the "Results and Discussion" section, only a simple numerical comparison of the performance parameters of this study (such as sensitivity) was made with just two or three scattered references, lacking in-depth discussion. For instance, the possible reasons for the performance differences (such as differences in sensing materials, reading distances, and circuit structures) were not analyzed.
Suggestion: It is recommended to establish a separate "Comparison and Discussion" section, systematically comparing the core performance indicators of this research (sensitivity, linearity, range, and reading distance) with 5 to 8 representative works published within the last five years, and conducting an in-depth discussion on the advantages and limitations of this scheme.
The quality and information content of the charts can be further improved.
Comments: The font size of the axis labels in Figure 4 (Sensor Frequency-Temperature Response Curve) is too small, and the data points in the figure are not clear. Figure 5 (Physical Picture) fails to clearly show the relative position and distance between the sensor and the reading antenna, which is crucial for readers to replicate the experiment.
Suggestion: Redraw all the charts to ensure that the axis labels and legends are clear and legible. Add a scale to the physical diagrams and it would be best to indicate the reading distance and experimental setup in the form of a schematic diagram.
The discussion on potential applications and limitations is too brief.
Opinion: The conclusion section of the paper merely mentions in a general way that it has application prospects in the biomedical and industrial fields, but does not discuss its applicability in specific scenarios. At the same time, it avoids addressing the obvious limitations of this design (such as the short reading distance and possible influence of metal environments).
Suggestion: In the discussion or conclusion section, add a paragraph on "Analysis of Application Scenarios and Limitations". Specifically explain the scenarios where this sensor is most suitable for application (such as implantation in the body, internal monitoring of sealed equipment), and candidly list the current design's shortcomings to provide suggestions for future research directions.
Reviewer 2 Report
Comments and Suggestions for Authors
With the efforts of the authors, the issues have been settled. All required questions have been answered.
Author Response
Thank you for your time and constructive feedback throughout the review. We appreciate your positive assessment and are glad that our revisions addressed all of your questions.
Reviewer 3 Report
Comments and Suggestions for Authors
The authors have answered all the questions and revised their manuscript accordingly.
Author Response

(The authors gave the same response as above.)

Reviewer 5 Report
Comments and Suggestions for Authors
The authors have addressed my questions and revised the paper accordingly. It is acceptable.
Author Response

(The authors gave the same response as above.)

Round 3
Reviewer 1 Report
Comments and Suggestions for Authors
Accept